# Who and Where Am I? Embodied Cognition-Aware Virtual Humans

## Abstract

Building virtual humans requires more than just realistic appearances and diverse motions; it necessitates simulating the intricate interplay between internal cognitive states, external environments, and executed motion behavior, as framed by the concept of embodied cognition. In this paper, we propose an embodied cognitive architecture, *EmbodiedHuman*, that captures this interaction by integrating "Mind" - a structured cognitive module, with motor execution to drive the virtual human's behavior within an interactive 3D environment. To enable integrated embodiment over both cognitive states and physical execution, we introduce three novel modules in a unified framework: 1) a cognition-inspired Mind structure, which models and modularize high-level reasoning and decision-making through key causal variables (value, belief, desire, and intention); 2) an action execution module, which translates internal intentions into embodied movements, enabling physically grounded interactions; and 3) an exploration module, which empowers the agent to actively explore the environment and update its mental states through feedback of actions. Our approach allows virtual humans to continuously adapt, learn, and evolve their behavior in response to environmental changes with autonomy, supporting dynamic and natural human-like interactions in the long horizon. Extensive experiments demonstrate the flexibility and scalability of our method in simulating individualized, daily-level behaviors in unknown environments. Project page: https://embodiedhuman.github.io.

## 1 Introduction

*What defines a truly "virtual human"?* Traditional research in this domain has primarily focused on creating realistic appearances (Kolotouros et al., 2024; Liu et al., 2024), replicating diverse human movements (Tevet et al., 2023; Zhu et al., 2024), and simulating them using fixed rules in simulation platforms (Puig et al., 2024). but these are not the whole picture. To create genuinely "human-like" virtual agents, we must simulate the *complexities* of *human behavior*, which arise from the interplay between internal mental states and external environments. A human-like agent requires cognitive abilities where the mind drives the body to interact with the environment, and in turn, these interactions shape the mind. This aligns with the philosophical notion of *embodied cognition*, where cognition is not merely a product of the brain but emerges through the agent's physical interactions with its environment (Clark, 1998; Wilson, 2002; Varela et al., 2017).

Nonetheless, to implement such a framework for virtual humans is far from straightforward. The primary challenge lies in simulating the complex, dynamic nature of an agent's internal mental states and their interplay with an external world that is ever-changing, only partially observable, and revealed incrementally through embodied interaction. Unlike traditional systems that focus on predefined, rule-based behaviors, a truly human-like agent must have autonomy across levels – adapting its actions on-the-fly based on evolving mental states which are shaped by its continuous sensory feedback from the environment.

To achieve this, we introduce an embodied cognitive architecture, *EmbodiedHuman*, that seamlessly integrates a high-level cognitive *"Mind"* concept with low-level motor execution (Fig. 1), allowing virtual humans to interact with environments based on their own mental and physical states.

Specifically, grounded in foundational theories in psychology and cognitive science (Bratman, 1987; Rao et al., 1995; Dennett, 1988; Wooldridge, 2003), we structure the *"Mind"* via modularization

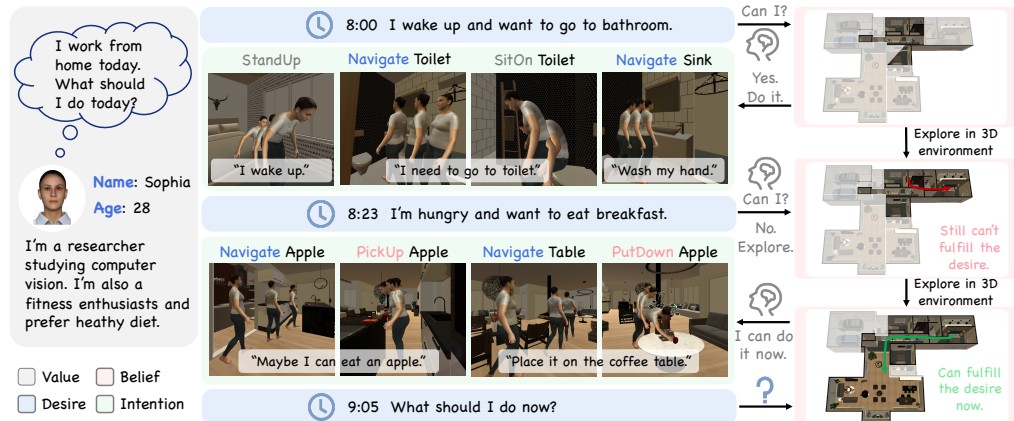

Figure 1: **EmbodiedHuman.** We propose an embodied cognition-aware framework to simulate natural daily behaviors of personalized virtual humans. Equipped with its own personality and preferences (*values*). When a virtual human is randomly placed into an unfamiliar environment, it continuously generates temporary *desires* that drive it to explore surroundings while updating its *beliefs* about the environment. Accordingly, the agent formulates specific *intentions* to fulfill its *desires*, which are further translated into embodied actions, such as navigation, static object interaction, movable object interaction, and free-form human motions.

through four causal variables: *value*, *belief*, *desire*, and *intention*, it offers a principled framework for self-directed reasoning and action in virtual humans. Concretely, within the *Mind*, *value* refers to an individual's intrinsic motivations, preferences, or long-term goals; *belief* reflects the agent's perception of the world, which may be accurate or inaccurate; *desire* encapsulates transient or context-dependent aspirations that influence behavior, distinguishing itself from long-term values; and *intention* is the specific action plan chosen to fulfill these *desires*, typically formed through a concrete decision-making process based on the agent's *beliefs* and *desires*. These causally interconnected elements collectively shape the agent's rich and adaptive behavior, as illustrated in Fig. 2.

While such a structured *Mind* concept grounds the internal cognitive states of the agent, key challenges remain: *1)* how can the agent physically enact its *intentions* through its *body*, interact with the *environment*, and *2)* in turn, allow these interactions to shape and update its mental states? To address this, we introduce two complementary interaction modules that operationalize the cognitive architecture, enabling bidirectional adaptation between cognition and embodied experience. The first is the *action execution* module, which translates the agent's *intentions* into a sequence of executable embodied actions.

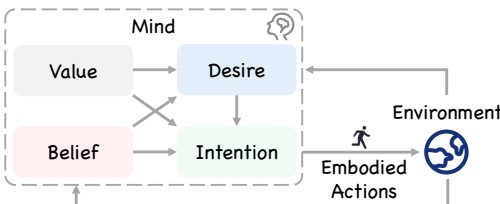

Figure 2: **Embodied cognitive architecture.** The conceptual workflow of proposed embodied cognitive architecture, shaping the *Mind* and its dynamic interaction with the *Environment* through embodied actions.

Leveraging large-scale human motion data (Guo et al., 2022), we develop a spatial-aware motion diffusion model to enable controllable motion generation with spatial control, supporting diverse and versatile actions that span multiple levels of granularity, including navigation, object interaction, and free-form human motions. The second is the *exploration* module, which enables the agent to purposefully explore its surroundings and continuously update its mental states (*i.e.*, *beliefs* and *desires*) based on environmental feedback. Specifically, we propose an exploration strategy that directs the agent towards unknown areas based on its current desires, the frontiers (Yamauchi, 1997) of the scene, and scene understanding.

To summarize, we propose to model virtual humans via an embodied cognition-aware framework, beyond separate motion synthesis or mind modeling. Our technical innovations can be concluded as follows: 1) We propose EmbodiedHuman, enabling the virtual human to continuously adapt, learn, and evolve its behavior based on its own characteristics and interactions with the environment. 2) We design an action execution module that leverages a spatial-aware motion diffusion model to achieve versatile and controllable motion generation, enabling rich environmental interactions. 3) We develop an exploration module that allows the agent to actively explore the environment, gather

information, and establish a feedback loop between its internal mind and the external world. 4) Experiments reveal the emergence of diverse, lifelike behaviors—such as picking its favorite foods, exercising, dancing, or simply relaxing in a room—just like any of us in our daily lives. We hope EmbodiedHuman can serve as a baseline prototype to advance research in embodied cognition and virtual human modeling. To support adoption and reproducibility, we will release upon acceptance a user-friendly framework with well-documented interfaces, modular components, and extensible APIs. It integrates all modules into a cohesive workflow, offering researchers and practitioners accessible tools to build, customize, and evaluate virtual humans at scale.

## 2 RELATED WORKS

**Human Behavior Simulation.** Human behavior simulation has been widely and long-term studied across various domains. Early works (Shao & Terzopoulos, 2005; Funge et al., 1999) simulate human behavior by defining rules for human-environment interactions and generating motion through basic motor skills. However, these approaches were limited to highly abstracted environments, and the rule-based methodology constrained both behavioral diversity and motion quality. For behavior planning, cognitive architectures like SOAR (Laird, 2019) and ACT-R (Anderson & Lebiere, 2014) model human-like decision-making and learning, while the BDI (Belief-Desire-Intention) model (Rao et al., 1995) simulates rational agents driven by cognitive states. In virtual environment platforms, VirtualHome (Puig et al., 2018) simulates 3D household tasks with atomic actions, and Watch-And-Help (Puig et al., 2020) enables collaborative behaviors. BEHAVIOR (Srivastava et al., 2022) offers a benchmark for realistic human activities, and Habitat 3.0 (Puig et al., 2024) provides an advanced simulation platform with accurate humanoid simulation and collaborative tasks. V-IRL (Yang et al., 2024) attempts to drive virtual humans to interact with realistic urban scenes, while it lacks physical embodiment. Our work advances these efforts by designing an embodied cognitive architecture that enables agents to autonomously explore, plan, generate and execute embodied actions in interactive environments, with a dynamic feedback loop that updates their mind, allowing diverse and adaptive behavior in unfamiliar environments.

**Conditional Motion Generation.** Generating human motion from various conditions has attracted widespread attention (Petrovich et al., 2022; Guo et al., 2022; Tevet et al., 2023; Chen et al., 2023; Zhang et al., 2023; Zhu et al., 2024; Wang et al., 2024a). Recent advances such as MDM (Tevet et al., 2023) leverage diffusion models for motion generation. MLD (Chen et al., 2023) proposes motion latent diffusion for better efficiency. Although achieving high-quality motion generation, prior methods typically employ relative motion representation, which undermines spatial controllability, as it lacks a sense of global positions. Several methods attempt to introduce spatial control based on MDM (Karunratanakul et al., 2023; Xie et al., 2024), while high control precision remains challenging under the relative representation. To address this, we propose a spatial-aware motion diffusion model to incorporate spatial information, thereby enhancing control precision.

**Human-Scene Interaction.** One line of works trains motion control policies through reinforcement learning (RL) and imitation learning to drive humanoid robot to navigate (Rempe et al., 2023; Cheng et al., 2025) and interact with environment (Tessler et al., 2024; Zhao et al., 2023; Peng et al., 2021; 2022; Fu et al., 2024). However, these approaches generally focus on short-term motion and are limited by predefined motion patterns or overfitting to specific environments, overlooking the cognitive nature of virtual humans. The other focuses on scene-aware motion generation (Hassan et al., 2021; Wang et al., 2022; Cen et al., 2024; Jiang et al., 2024b;a; Wang et al., 2024b; Pi et al., 2023). For example, TRUMANS (Jiang et al., 2024b) expands human-scene interaction dataset and trains an autoregressive motion diffusion model to generate motion based on action label. LINGO (Jiang et al., 2024a) further extends TRUMANS by incorporating textual descriptions into the dataset. Despite remarkable progress, previous approaches generally assume a *known* environment and generate separate motion sequences. Moreover, they synthesizes human motions primarily based on external instructions, overlooking the internal mental states of virtual humans.

## 3 METHODOLOGY

In this section, we first present the high-level embodied cognitive architecture (Sec. 3.1 and Fig 3), followed by details of the *mind* (Sec. 3.2) and the two interaction modules (Sec. 3.3).

Figure 3: **Framework of our EmbodiedHuman.** To simulate a truly "virtual human" in an unfamiliar environment (Sec. 3.1), we design an embodied cognitive architecture, *EmbodiedHuman*, shaping the mind with *value*, *belief*, *desire*, and *intention* (Sec. 3.2), and coupling cognition with actions for embodied interaction. Starting from the *value* (virtual human profile) and the *belief* (dynamic scene graph), we leverage an LLM to generate temporary *desires*, which are translated into concrete *intentions*. Two interactive modules are proposed to enable embodied *action execution* and dynamic environment *exploration* (Sec. 3.3).

## 3.1 EMBODIED COGNITIVE ARCHITECTURE

To define a virtual human's behavior, we consider a triplet $(\mathcal{M}_t, \mathcal{A}_t, \mathcal{E}_t)$, where $t$ is the simulated time. $\mathcal{M}_t$ represents the *Mind* at time $t$, *i.e.*, the internal cognitive states, including *value*, *belief*, *desire*, and *intention*; $\mathcal{A}_t$ represents the *Action* taken by the agent at time $t$, reflecting its embodied interaction with the environment; and $\mathcal{E}_t$ represents the *Environment* at time $t$, *i.e.*, the external world that the virtual human perceives and interacts with. Simulated time $t$ can be associated with real-world time (*e.g.,* 8:30 AM) when performing daily activities. See *Appendix* A.4 for more details.

We formulate the interaction between the three components via the following relationships inspired by *embodied cognition*, which emphasizes that cognitive processes emerge from continuous interactions between the mind, body, and environment: (1) The *Action* $\mathcal{A}_t$ of the virtual human is directly determined by its internal cognitive state, $\mathcal{M}_t$, and the external environment, $\mathcal{E}_t$, via $\mathcal{A}_t = \pi(\mathcal{M}_t, \mathcal{E}_t)$, where $\pi$ is the policy function that maps $\mathcal{M}_t$ and $\mathcal{E}_t$ to a specific action, reflecting the influence of both internal mind and external world on the behavior. (2) The *Environment* $\mathcal{E}$ evolves as a result of the action. Specifically, the new environmental state $\mathcal{E}_{t+1}$ is determined by the previous environment $\mathcal{E}_t$ and the action $\mathcal{A}_t$. Formally, $\mathcal{E}_{t+1} = \mathcal{T}(\mathcal{E}_t, \mathcal{A}_t)$, where $\mathcal{T}$ models how the environment changes in response to actions. (3) The *Mind* $\mathcal{M}_{t+1}$ is updated based on the outcome of the action $\mathcal{A}_t$ interacting with the environment $\mathcal{E}_t$, *i.e.*, $\mathcal{M}_{t+1} = \psi(\mathcal{M}_t, \mathcal{A}_t, \mathcal{E}_t)$, where $\psi$ is a function that models the evolution of the mental states, incorporating feedback from the actions taken and changes in the environment.

These relationships model the dynamic interaction and feedback loop between the virtual human's mind, body (actions), and environment, reflecting key insights of *embodied cognition*. Concretely, the environment transition function $\mathcal{T}$ represents the objective evolution of the external environment, which is typically specified by the environment itself. Our primary focus, therefore, is on the implementation of the policy function $\pi$ and the mind update function $\psi$ to more effectively simulate dynamic human behavior through the interplay of the mind, body, and environment.

## 3.2 MIND

*Mind* consists of four interconnected modules (*value*, *belief*, *intention* and *desire*), structured through causal relationships that model human decision-making process. Below, we outline the key design principles, while additional details can be found in the *Appendix* C.

**Value.** The virtual human's *value* ($V$) represents its intrinsic motivations, preferences, and goals. These *values* remain relatively stable over the timescale we study, typically measured in hours or days, and guide the agent's overall behavior in the environment. We define the virtual human's *value* via a character profile, describing attributes such as profession, personality, and other characteristics.

**Belief.** The agent's *belief* ($B_t$) represents its perception of the environment at time $t$. The *belief* is updated based on observations the agent makes as it interacts with the environment. In our framework, **the agent does not have access to oracle-like global scene information**; instead, it **perceives**

**the environment from an egocentric view** and dynamically forms its *belief* through continuous interaction. Thus, *belief* may be accurate or biased at any given moment, but is gradually refined over time as the agent gathers more information from its surroundings.

To represent this evolving *belief*, we employ a dynamically updated 3D scene graph composed of the objects the agent has explored. Each object is characterized by properties such as its 3D location, geometric information, and optional affordances, as illustrated in Fig. 3. More details are provide in *Appendix*. To achieve open-world perception in unknown environments, we employ Grounded-SAM (Ren et al., 2024; Kirillov et al., 2023; Liu et al., 2023) to extract semantic information from egocentric RGB observations, which is projected into 3D space using depth to generate 3D instance-level point clouds. As the virtual human interacts with the environment, the scene graph is progressively updated. This process can be formulated as:

$$B_{t+1} = \mathcal{F}_B(B_t, \mathcal{E}_{t+1}), \tag{1}$$

where $\mathcal{F}_B$ denotes the belief update function, which is implemented by estimating the overlap between newly perceived instances and existing ones, and then merging them accordingly.

**Desire.** The agent's *desire* ($D_t$) reflects its immediate, context-driven goals at time $t$. Each *desire* can be mapped to a scheduled conceptual time to enable human-readable time representation. While *value* represents long-term motivations, *desire* is more specific and dynamic, shaped by the agent's current situation and needs. For example, a *desire* might manifest as "*I want to grab some food*" or "*I want to take a break*" at the moment. The generation of a particular *desire* is closely influenced by the agent's *value* and evolving *belief* about the environment, which can be expressed as:

$$D_t = \mathcal{F}_D(V, B_t). \tag{2}$$

**Intention.** The agent's *intention* ($I_t$) represents a concrete, actionable plan formed to achieve its goals at time $t$. While *desire* expresses what the agent wants (*e.g.*, "*I want to eat*"), *intention* addresses the question of how the agent intends to fulfill that *desire* (*e.g.*, "*I will go to the kitchen and cook a steak*"). In this way, *intention* serves as a bridge between *desire* and action, translating abstract goals into specific steps. Shaped by *value*, *desire*, and the current environmental understanding (*i.e.*, *belief*), this process can be formulated as:

$$I_t = \mathcal{F}_I(V, D_t, B_t). \tag{3}$$

### 3.3 INTERACTION MODULES

We propose two interaction modules to enable the agent to execute physically grounded *Actions* in the environment while establishing a dynamic feedback loop between its internal *Mind* and the external *Environment*. Specifically, if the agent determines, based on its current *belief*, that its *desire* can be satisfied, the *action execution module* translates *intentions* into embodied actions. Otherwise, the *exploration module* actively drives the agent to explore the environment, guided by the agent's *belief* and *desire*, and, in turn, updates them as new information is gathered from the surroundings.

#### 3.3.1 ACTION EXECUTION

The action execution module translates *intentions* into executable humanoid actions. For precise scene interaction and diverse motion simulation, it requires two key properties: precise spatial motion controllability and free-form motion generation.

While employing RL to control humanoid actions is a straightforward solution, it often yields limited and unnatural motion patterns and can easily overfit to specific scenes. Therefore, leveraging motion generation with priors from large-scale human motion data offers a more promising path, either via standalone (Tevet et al., 2023) or scene-aware (Jiang et al., 2024b) manners.

However, such pre-trained models often fail to satisfy both properties—either lacking effective environmental interaction or struggling to generalize across diverse environments due to data limitations. Based on the above observations, we develop a spatial-aware motion diffusion model (SA-MDM), which enhances spatial control capability through two design levels: *spatial-aware motion representation* and *gradient-based motion control*. Such designs improve the virtual human's ability to interact with the environment while utilizing large-scale scene-agnostic human motion data. As shown in Fig. 4, SA-MDM enables precise motion control via both texts and spatial cues, allowing the virtual human to perform specific, context-aware actions. We unfold the details below.

**SA-MDM.** Most previous human motion generation models follows the relative, local motion representation proposed in HumanML3D, meaning that the joint representation of the current frame is relative to the previous frame. This representation diminishes the ability for precise global position control (Karunratanakul et al., 2023), which is crucial for ensuring physically plausible interactions with the environment. To address this, we primarily use global joint positions $p$ as the motion representation. However, this representation is limited to skeleton-level control and does not account for the virtual human's physical volume. Consequently, relying solely on joint control is inadequate for preventing penetration with surrounding objects. Thus, we propose a hybrid spatial-aware motion representation that fuses joint positions with SMPL-X parameters $\theta$.

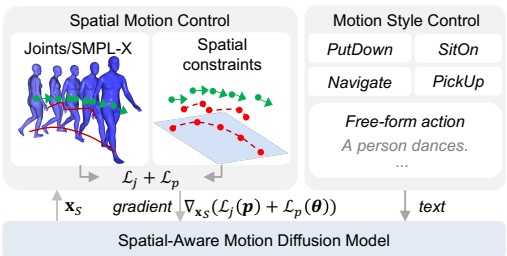

Figure 4: **Overview of the action execution module.** Text controls the motion style of Spatial-Aware Motion Diffusion Model (SA-MDM). The diffusion process is further guided by gradients from spatial constraints, where red curves indicate sparse joint locations (taking root and hand joints as examples) and green arrays are global orientations.

We follow MLD to perform diffusion in a latent space (compressed with VAE) for better efficiency and motion quality. The two representations are tightly coupled throughout the generation process to ensure coherence, where joints govern the global position and posture, while the mesh vertices derived from SMPL-X parameters provide the necessary body shape information for penetration avoidance.

**Motion Control.** With SA-MDM, we can control the motion in a spatial-aware manner through two key components. First, we regulate the global joint positions to perform specific environment-aligned actions by designing tailored joint loss functions $\mathcal{L}_j$ for different types of controls (*e.g.*, joint position, heading angle, *etc*). Second, we reduce potential human–scene penetration by computing a penetration loss $\mathcal{L}_p$ between mesh vertices and the dynamically updated scene occupancy. Thanks to our spatial-aware motion representation, both forms of control can be seamlessly applied by incorporating the guidance directly into the denoised latent code, which is formally expressed as:

$$\tilde{\mathbf{x}}_s = \mathbf{x}_s - \alpha \nabla_{\mathbf{x}_s}(\mathcal{L}_j(\boldsymbol{p}) + \mathcal{L}_p(\boldsymbol{\theta})), (\boldsymbol{p}, \boldsymbol{\theta}) = \mathcal{D}(\mathbf{x}_s) \tag{4}$$

where $\mathcal{D}(\cdot)$ represents the decoder of the VAE in SA-MDM, $\tilde{\mathbf{x}}_s$ and $\mathbf{x}_s$ denote the perturbed and original latent estimates at time $s$, and $\alpha$ represents the gradient scale, respectively. Using this strategy, we can effectively control virtual humans' interaction with the external environment in a unified manner. Additionally, we enable text-based motion control, allowing for freeform style adjustments. These texts leverage the agent's cognitive context, making the motion more flexible and personalized. More details are provided in the *Appendix* A.4.

**Training with Multi-Source Data.** To enhance motion quality for complex interactions, we augment the HumanML3D (Guo et al., 2022) dataset with additional human-object interaction datasets, including SAMP dataset (Hassan et al., 2021) and GRAB dataset (Taheri et al., 2020).

**Hand-object Alignment.** To create more realistic hand poses during object interaction, we leverage GRIP (Taheri et al., 2024) to generate hand poses based on hand and object geometry. Since GRIP does not provide object's poses, we determine it via a two-step process: first, we perform an initial alignment using the hand's surface normals, and second, we track the object's pose frame-by-frame with hand's rigid transformation by registering the hand vertices. See *Appendix* D for visualization.

### 3.3.2 EXPLORATION

Notably, since the agent dynamically gathers information and forms its *belief* about the environment through continuous interactions (as discussed in Sec. 3.2), it may not always be able to form a specific *intention* that fulfills its *desire* (*e.g.*, when a required object is in another room, and the agent is unaware of it). In such cases, we propose an exploration module with a desire-driven strategy, where exploration itself becomes a form of *intention*. This enables the agent to actively seek new information in response to its unmet *desire*, and further update its mental states.

Specifically, the agent needs to distinguish between known and unknown areas, and identify regions for further exploration. In practice, we compute two distinct maps during exploration: explored

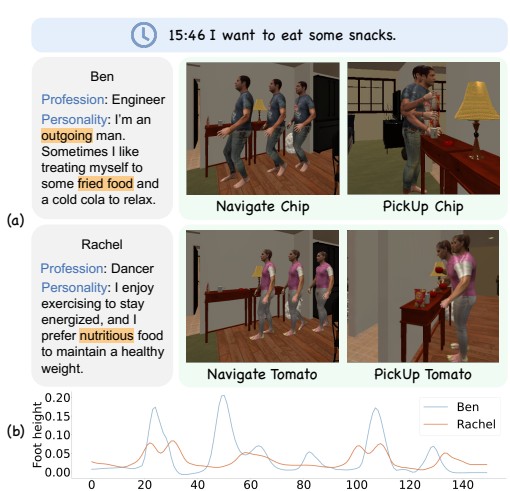

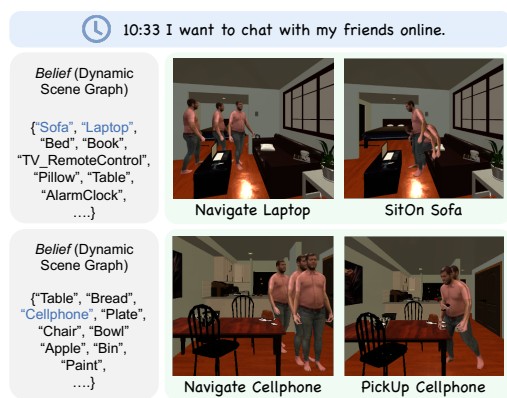

Figure 5: *Value* **influences virtual human behavior.** (a) Two virtual humans (*i.e.*, Ben and Rachel) tend to eat different foods (Chip or Tomato) according to their *values*. (b) Moreover, as an outgoing individual, Ben's gait appears brisker, which is also reflected in the foot height visualization at the bottom.

Figure 6: *Environment* **influences virtual human behavior.** Armin chooses different assets (Laptop or Cellphone) to fulfill his *desire* based on his *belief* about the environment.

| Method | Daily Plan. ↑ | Desire Plan. ↑ |
|---|---|---|
| w/o Exploration | 0.05 | 0.54 |
| w/o Desire | 0.45 | 0.91 |
| Ours | **0.65** | **0.95** |

Table 1: Evaluation of the exploration module.

map and obstacle map. By analyzing their differences, we detect frontiers (*i.e.*, boundary regions between explored and unexplored spaces) (Yamauchi, 1997). However, these frontiers lack semantic information. To bridge this gap and connect the frontiers with human-like cognition, we employ an LLM to assess the exploration priority of each frontier based on the current *desire* and descriptions of surrounding object instances. For an intuitive example, if the current *desire* is to go to the bathroom, a frontier near a sink would more likely be useful and therefore prioritized for exploration. Nonetheless, there are times when the *desire* simply cannot be fulfilled within the environment. We terminate the process when no frontiers remain or when the exploration for a single *desire* exceeds a certain number of attempts. At that point, we update the agent's *desire* based on the feedback, just as a real human would. This process reflects how active interaction with the environment influences the virtual human's mind, including its *belief* and *desire*. Refer to *Appendix* C for details.

## 4 EXPERIMENTS

In this section, we first brief describe our experimental setup (Sec. 4.1), then present the main results of the daily human behavior simulation (Sec. 4.2). Finally, we evaluate the two main interactive modules, *i.e.*, action execution (Sec. 4.4) and exploration (Sec. 4.3) .

### 4.1 EXPERIMENTAL SETUP

We model $V$, $B_t$, $D_t$, and $I_t$ using natural language and implement $\mathcal{F}_D$ and $\mathcal{F}_I$ by prompting a pre-trained LLM (GPT-4o (Hurst et al., 2024) in our experiments), considering the complexity of human mental states. We adopt Habitat 3.0 (Puig et al., 2024) as the simulation platform and use SMPL-X (Pavlakos et al., 2019) as the skeleton representation. We select 10 scenes from HSSD (Khanna et al., 2024) and AI2-THOR dataset (Kolve et al., 2017). We define 10 virtual human profiles encompassing different professions, personalities, daily goals, *etc*. We then randomly assign virtual humans to scenes, creating 20 human-scene combinations. Refer to *Appendix* for more details.

### 4.2 DAILY HUMAN BEHAVIOR SIMULATION

We investigate two key research questions (RQs) in this subsection to examine if EmbodiedHuman can reflect variations in internal *value* and external environment, and generate diverse, contextually appropriate behaviors at the meantime:

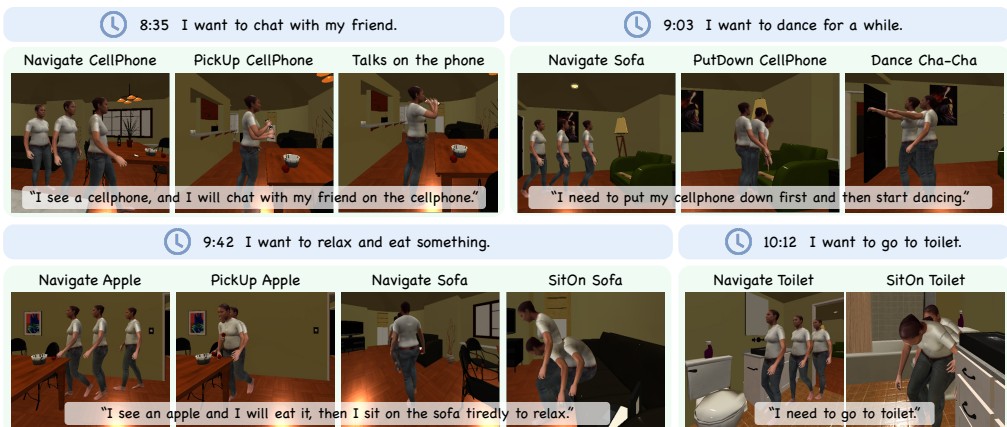

Figure 7: **A day of Fiona.** Fiona is a manager working at a trading company. She is a social butterfly and is passionate about dancing. Today is a holiday. We generate long-term daily behaviors of Fiona: She starts the day by chatting with her friend on the phone and then dances to wake up herself. Thereafter, she feels tired and sits down to eat an apple. Then, she needs to go to toilet. The simulated behaviors reflect the *value* of Fiona and the *belief* from the environment.

**RQ1: How does *value* influence virtual human behavior?** Virtual human behavior is shaped by their *values*, leading to distinct choices even in the same environment. To examine this, we conduct a controlled experiment with two virtual humans sharing the same *desires* in the same environment. As shown in Fig. 5(a), Ben and Rachel select different foods according to their dietary preferences: Ben chooses chips due to his preference for fried food, while Rachel opts for a tomato, valuing her nutritional benefits. Additionally, we compare the quantitative gait difference via visualizing the foot height of both virtual humans (Fig. 5(b)). We observe that Ben's gait is brisker, aligning with his *value* of being "outgoing."

**RQ2: How does the *environment* influence virtual human behavior?** The external environment plays a crucial role in shaping the agent's cognition and behavior. As shown in Fig. 6, we place the virtual human Armin into two different environments with the same *desire* and observe different decisions. To fulfill the *desire* to "chat with friends online," Armin chooses either a cellphone or a laptop, depending on which is available in the environment. By perceiving environmental cues, the agent forms different *beliefs*, leading to diverse yet reasonable behaviors.

**Simulate a Day of Virtual Humans.** Based on the ability of EmbodiedHuman, we can simulate dynamic and diverse human behaviors. Figs. 7 and 8 depict a simulated day in the lives of two virtual humans, Fiona and Diego. They have distinct professions and personalities, which lead them to make different choices throughout the day. For example, as a social butterfly, Fiona enjoys chatting on the phone and dancing, while Diego, an athlete, prioritizes working out to maintain his energy. The results show that our pipeline can generate a wide range of human behaviors based on the virtual humans' *values*, allowing for rich and meaningful interactions with the environment. This is achieved through the effective interaction between the cognition and the environment, along with diverse human actions, including navigation, static object interactions (*e.g.*, *SitOn*), dynamic object interactions (*e.g.*, *PickUp*), and open-vocabulary human motions (*e.g.*, *Dance Cha-Cha*).

## 4.3 EVALUATION OF EXPLORATION MODULE

We investigate the exploration module from two aspects: (1) success rate (in main manuscript), and (2) efficiency (*Appendix* D). Concretely, we introduce two metrics to assess the planner's ability to generate valid *intentions* that fulfill *desires*: Desire Plan Success Rate (*Desire Plan.*) and Daily Plan Success Rate (*Daily Plan.*). The former measures the ratio of individual *desires* that can be fulfilled based on the agent's *belief*, while the latter evaluates the success rate of fulfilling all *desires* within a day. We experiment with removing *desire* guidance and ablating the exploration module entirely. Table 1 shows that both *Daily Plan.* and *Desire Plan.* drop significantly without exploration. In this scenario, the virtual human's ability to achieve its *desires* depends solely on its initial position; if the current *belief* cannot satisfy the *desire*, the plan fails. Additionally, incorporating *desire* further improves success rates by enabling more efficient, semantically-aware exploration.

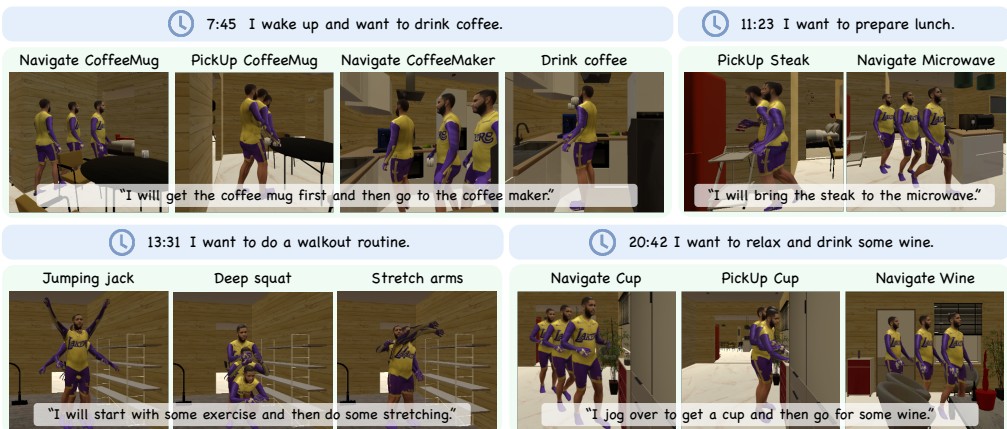

Figure 8: **A day of Diego.** As an athlete, he enjoys drinking coffee to stay alert and occasionally has alcohol to relax. Today, he needs to exercise. He starts the day with a cup of coffee to energize himself. As he prioritizes high-protein foods, he chooses steak for lunch. Afterward, he completes a workout routine to meet his daily goal. In the evening, he treats himself to a glass of wine to unwind.

## 4.4 EVALUATION OF ACTION EXECUTION MODULE

We assess the precision of the proposed action execution module using the following metrics: Daily Executed Rate (*Daily Exec.*), Desire Executed Rate (*Desire Exec.*), Action Executed Rate (*Action Exec.*), Goal Distance (*Goal Dist.*), and Penetration Ratio (*Pene.*). Specifically, *Goal Dist.* refers to the distance between the joint and the goal; *Action Exec.* measures the rate at which the executed action successfully realizes the intention (*i.e.*, when *Goal Dist.* is below a predefined threshold $\tau$). *Daily Exec.* and *Desire Exec.* represent the proportion of fully successful intentions at the daily and desire levels. For example, *Daily Exec.* requires all intentions within a day to succeed; if any fail, the entire day is considered unsuccessful. Please refer to the *Appendix* B, D for more details and analysis of SA-MDM.

Table 2: **Evaluation of action execution module.** [*]We adapt the baseline methods to our framework by carefully modifying them to replace our action execution module.

| Methods | Daily Exec.↑ | Desire Exec.↑ | Action Exec.↑ | Goal Dist.↓ | Pene.↓ |
|---|---|---|---|---|---|
| OmniControl[*] (Xie et al., 2024) | 0.05 | 0.64 | 0.83 | 0.313 | 0.039 |
| TextSceneMotion[*] (Cen et al., 2024) | 0.00 | 0.57 | 0.80 | 0.265 | 0.209 |
| TRUMANS[*] (Jiang et al., 2024b) | 0.20 | 0.70 | 0.88 | 0.267 | 0.033 |
| Ours | **0.30** | **0.79** | **0.91** | **0.237** | **0.027** |

**Comparison with SOTA.** We compare our method against two scene-aware human motion generation methods (*i.e.*, TextSceneMotion (Cen et al., 2024) and TRUMANS (Jiang et al., 2024b)) and a spatially controllable motion generation method (*i.e.*, OmniControl (Xie et al., 2024)). Specifically, we replace our action execution module with each baseline method, adapting them to fit our scenarios. As shown in Table 2, TextSceneMotion exhibits a notably higher penetration ratio, likely due to its object-centric approach. However, limitations in its dataset hinder its performance when transferred to new environments in Habitat 3.0. TRUMANS benefits from a larger human-scene interaction dataset, but its success rates remain significantly lower than ours, especially at the desire and daily levels, where errors accumulate across actions. OmniControl employs joint position control for human-scene interaction, but its precision is lower than ours, leading to much lower success rates. We attribute this to its reliance on relative motion representation.

## 5 CONCLUSION

We propose EmbodiedHuman, an embodied cognitive architecture to simulate realistic virtual human behavior by integrating "Mind" with two interactive modules. We shape the "Mind" with four interconnected causal variables: *value*, *belief*, *desire* and *intention*. To achieve physically grounded interactions, we propose an action execution module to generate diverse motions. We further design a desire-driven exploration module to actively explore unknown environments. Experiments show that EmbodiedHuman supports individualized, daily-level behavior simulation.

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

## APPENDIX

In this appendix, we provide: (1) additional implementation details (Section A); (2) benchmark details (Section B); (3) an extended discussion on the cognitive modeling of the Mind (Section C); (4) further experimental results and analysis (Section D); and (5) a discussion on the broader impact/limitations/future work of our method (Section E). Additionally, we provide a *Supplementary Video* showcasing key design elements.

## CONTENTS OF APPENDIX

## A   IMPLEMENTATION DETAILS

### A.1   PROBLEM DEFINITION

This work aims to simulate embodied virtual human behaviors in an unfamiliar environment. The input includes the *value* of a virtual human and an unexplored environment. The output are human behaviors, which are represented with human motions and their interactions with the environment.

### A.2   TRAINING DETAILS

We conduct all experiments on a single NVIDIA L40 GPU. The latent code dimension of SA-MDM is set to $1 \times 256$, following the original MLD (Chen et al., 2023). For denoising diffusion, we employ DDIM (Song et al., 2021) as the scheduler, setting the number of denoising steps to 200.

### A.3 MOTION CONTROL

As mentioned in the main manuscript, we apply guidance to SA-MDM to control the human motion. Our joint control signals consist of two types: joint position and heading angle. To control the joint positions of the virtual human, we calculate the $L_2$ distance between the predicted joint position and the target position. To control the heading angle of the virtual human, we estimate the heading angle based on the joint positions and then compute the $L_1$ distance between them.

We observe that applying guidance to all diffusion steps leads to a significant decrease in quality and perturbing only a few diffusion steps can precisely control the joint position. Therefore, we only apply guidance at the last several diffusion steps. Additionally, we improve the control accuracy by repeating the guidance $K$ times within a single step with a small gradient scale $\alpha$. We set $\alpha$ to $0.05$ and $K$ to $50$ in our implementation. The guidance is applied to the last $5\%$ diffusion steps.

### A.4 IMPLEMENTATION IN SIMULATOR

We employ Habitat 3.0 (Puig et al., 2024) as the simulator, which supports humanoid agent control based on SMPL-X parameters. We use Grounded-SAM (Ren et al., 2024) for open-world object detection, which supports object category inputs. The detected objects are fused based on point cloud overlap and mapped to the objects in the simulator by calculating their center distances. The daily activity of the virtual human is discretized into a set of *desires*, each associated with a conceptual time of a day. Each desire is then transformed into a set of *intentions*, with each intention corresponding to a specific action. For each action, we use SA-MDM to generate a sequence of human motions for motion control in the simulator. To achieve interaction between the virtual human and the environment, we employ spatial control as discussed above. To ensure continuity between motions, the initial pose of each motion is regularized based on the final pose of the previous motion. We provide the details below:

(1) For "*PickUp*" action, we first grab the object by regularizing the wrist joint's distance to the object in the final frames and then use the final grabbing pose as the initial pose for picking-up. We also regularize the orientation of the virtual human to ensure it faces the object. The loss can be formulated as:

$$\mathcal{L}_j^{\text{pick}} = \|\mathbf{p}_w^{F_t} - \mathbf{o}\|_2^2 + \|\theta^{F_t} - \theta\|_2^2 + \sum_i^{22} \|\mathbf{p}_i^0 - \mathbf{p}_i^{F_{t-1}}\|_2^2, \tag{5}$$

where $\mathbf{p}_w^{F_t}$ represents the wrist joint's position in the last frame ($F_t$) of the current motion, $\mathbf{o}$ denotes the object position, $\theta^{F_t}$ is the orientation vector of the virtual human, $\theta$ is the target orientation vector to face the object, $\mathbf{p}_i^0$ represents the joint position in the starting frame of the current motion, $\mathbf{p}_i^{F_{t-1}}$ is the joint position in the last frame ($F_{t-1}$) of the previous motion.

(2) For "*PutDown*" action, we regularize the wrist joint's distance to a randomly generated position on the target object. The loss can be formulated as:

$$\mathcal{L}_j^{\text{put}} = \|\mathbf{p}_w^{F_t} - \mathbf{o}_t\|_2^2 + \sum_i^{22} \|\mathbf{p}_i^0 - \mathbf{p}_i^{F_{t-1}}\|_2^2, \tag{6}$$

where $\mathbf{o}_t$ is the randomly generated position on the target object.

(3) For "*Navigate*" action, we regularize the pelvis position to follow the planned trajectory. The loss can be formulated as:

$$\mathcal{L}_j^{\text{nav}} = \sum_f^{F_t/h} \|\mathbf{p}_p^{fh} - \mathbf{p}_{\text{traj}}^{fh}\|_2^2 + \sum_i^{22} \|\mathbf{p}_i^0 - \mathbf{p}_i^{F_{t-1}}\|_2^2, \tag{7}$$

where $h$ is the time interval for loss calculation, $\mathbf{p}_p^{fh}$ and $\mathbf{p}_{\text{traj}}^{fh}$ denote the pelvis position and trajectory point in frame $fh$, respectively.

(4) For "*Open*" and "*Close*" actions, we regularize the wrist joint trajectory with the estimated handle trajectory. The loss can be formulated as:

$$\mathcal{L}_j^{\text{o/c}} = \sum_f^{F_t/h} \|\mathbf{p}_w^{fh} - \mathbf{p}_{\text{traj}_h}^{fh}\|_2^2 + \sum_i^{22} \|\mathbf{p}_i^0 - \mathbf{p}_i^{F_{t-1}}\|_2^2, \tag{8}$$

where $\mathbf{p}_w^{fh}$ and $\mathbf{p}_{\mathrm{traj_h}}^{fh}$ denote the wrist joint's position and handle trajectory point in frame $fh$.

(5) For actions like "*SitOn*", "*LieOn*" and "*StandUp*", we directly generate the motion conditioned on the geometry of the target object.

(6) For "*DoIt*" action, we directly generate the motion based on the textual description.

To alleviate penetration with surroundings, we derive the mesh of virtual humans from the SMPL-X parameters $\boldsymbol{\theta}$. Denoting the vertices as $\mathbf{V} = \{\mathbf{v}_i\}_{i=1,\cdots,N_v}$, where $N_v$ is the number of vertices, the penetration loss $\mathcal{L}_p$ can be defined as:

$$\mathcal{L}_p = \sum_{i=1}^{N_v} \mathbf{O}(\mathbf{v}_i), \tag{9}$$

where $\mathbf{O}(\cdot)$ represents the occupancy map of the 3D scene, which is constructed online as the virtual human explores the environment.

## B    BENCHMARK DETAILS

### B.1    DETAILS OF EVALUATION

We employ two types of metrics for evaluation, *i.e.*, action-level metrics to evaluate the action execution module, and plan-level metrics to evaluate the exploration module. The action-level metrics include *Daily Exec.*, *Desire Exec.*, *Action Exec.*, *Goal Dist.*, and *Pene.*. The executed rates across three levels are defined based on *Goal Dist.*, *i.e.*, an action is considered as executed if *Goal Dist.* is below a threshold $\tau$, where $\tau$ is set to $0.5$m in our implementation. To evaluate the *Goal Dist.*, for whole-body movements such as *Navigation*, we calculate the closest distance between all joints to the goal object; for movements based on the hand (*e.g.*, *PickUp*), *Goal Dist.* is measured as the distance between the wrist joint and the goal object.

We simulate 20 human-scene combinations (*i.e.*, 20 days, 155 desires and 464 intentions). In general, an intention covers 1 to 3 action executions, corresponding to 80 to 500 frames for each.

### B.2    VIRTUAL HUMAN CONSTRUCTION

An individual virtual human consists of its *value* and appearance. To generate the *value*, we prompt the LLM to generate the name, profession, age, personality, gender and also daily goal. For appearance modeling, we utilize the texture maps from SMPLitex (Casas & Comino-Trinidad, 2023), ensuring that each virtual human asset is visually distinct, while maintaining alignment with personality traits. We employ the official Habitat 3.0 tool to convert these texture maps into textured SMPL-X models, which are represented using URDF (Unified Robot Description Format) models for seamless integration with physics-based simulation frameworks. This allows for realistic articulation, grasping, object manipulation, and full-body motion simulation, supporting scenarios where dexterous hand interactions are necessary.

### B.3    SCENE CONSTRUCTION

Habitat 3.0 provides various simulation environments. To meet the requirements for realistic and meaningful agent-environment interactions, we curate synthetic scenes for our benchmark from AI2THOR and HSSD, ensuring the diversity of interactive household objects, navigable spaces, and spatial layouts that could support a wide range of human-like activities. Additionally, to expand the diversity of object assets, we integrate high-quality 3D models from Sketchfab (Authors), including items such as *wine* and *beef*.

## C    ANALYSIS OF MIND

The *mind* is implemented using LLM (GPT-4o in our implementation). We summarize the implementation in Fig. A-1.

Table A-1: Example of generated *desires*.

> Example of generated desires.
>
> {
> "Desire1": { "Time":"07:30", "Desire":"Wake up and stretch.", },
> "Desire2": { "Time":"08:30", "Desire":"Prepare a healthy breakfast.", },
> "Desire3": { "Time":"09:00", "Desire":"Sit at my desk and start my work.", },
> "Desire4": { "Time":"12:00", "Desire":"Take a short break and have a nutritious snack.", },
> "Desire5": { "Time":"12:15", "Desire":"Try a new stretching routine to stay active.", },
> "Desire6": { "Time":"13:00", "Desire":"Take a cup of coffee.", },
> "Desire7": { "Time":"13:30", "Desire":"Continue working on my research at the desk.", },
> "Desire8": { "Time":"17:00", "Desire":"Make a healthy dinner.", },
> "Desire9": { "Time":"19:00", "Desire":"Sit down and watch TV.", }
> }

**Value.** *Value* represents the internal properties of the virtual human, including the profession, age, personality, daily goal, *etc*. *Value* influences the decision-making process of a virtual human, including the short-term *desire* and concrete *intention*, as illustrated in Fig. 4 of the main manuscript. We implement *value* using human profiles. As shown in Tables A-9 and A-10, we define 10 virtual humans in our experiments.

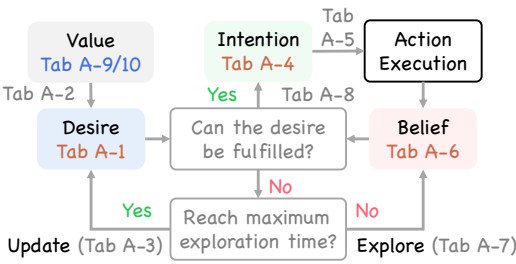

Figure A-1: Implementation of *Mind*. We provide examples and prompt designs for each *mind* component.

**Desire.** *Desire* represents the short-term goal of a virtual human. We implement *desire* by maintaining a sorted list, with each element containing a specific goal and a conceptual time. We provide examples of desires in Tab A-1. As discussed in the main manuscript, *desire* can be generated and updated based on the *value* and *belief*: *(1) Desire generation.* As shown in Table A-2, the LLM is prompted to generate *desires* of the virtual human according to its profile. To ensure feasibility, we impose constraints of supported actions that prevent the generation of *desires* that the action execution module cannot fulfill. *(2) Desire updating.* As discussed in the main manuscript, exploration is terminated either when no unexplored frontiers remain or when a single *desire* exceeds a predefined number of attempts. In such cases, the LLM updates the virtual human's *desire* by generating a new one based on prior *desires*, the virtual human's profile, and the current scene graph, as shown in Table A-3.

**Intention.** *Intention* is generated based on the short-term *desire*, the long-term *value*, and the *belief*. We implement intentions with a sorted list, where each element consists of an action to be executed and the object to be interacted with. We provide examples of *intentions* and their corresponding *desires* in Table A-4. We define several common *intentions* that can be completed with specific actions (*i.e.*, *PickUp*, *Navigate*). For free-form motions (*i.e.*, "dance") that do not require interaction with the environment, we assign *DoIt* as the *intention* and directly use text descriptions to control the action. *Intention* can then be transformed into human motions using SA-MDM. For more details, please refer to *Appendix* A.4.

We provide the implementation details for *intention* generation in Table A-8. The LLM is prompted to generate specific *intentions* based on the current *desire*, the virtual human profile, the current scene graph, and previous *intentions*. Besides, we prompt LLM to generate a motion style for whole-body movements based on the current and previous *intentions*, as shown in Table A-5.

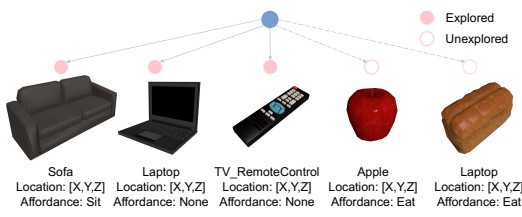

Figure A-2: Illustration of 3D scene graph.

**Belief.** *Belief* defines the perception results of the environment. As shown in Fig. A-2, we represent the *belief* as a scene graph, comprising a set

Table A-2: Prompts for *desire* generation.

> **Prompts for desire generation.**
>
> You are an virtual human in a room. Below is your profile:
> {VIRTUAL HUMAN PROFILE}
> How will you spend your day in the room?
> Please list desires of your day.
> The desires must can be completed using the following actions or completed by your self (do not require interactions with the environment):
> [1] Open: open some articulated objects.
> [2] Close: close some articulated objects.
> [3] PickUp: pick up something, can be used for eat something or drink something.
> [4] PutDown: put down something on something.
> [5] Navigate: go to somewhere in the room.
> [6] SitOn: sit on a chair or a sofa.
> [7] LieOn: lie on a bed.
> [8] StandUp: stand up from a chair, a sofa or a bed.
> [9] DoIt: if the desire does not need interaction with the environment, just plan as DoIt.
> Example desires:
> [1] I am hungry and want to eat something.
> [2] Someone knocks the door and I need to go to open the door.
> [3] I want to sit to watch TV.
> [4] I want to go to toilet.
> The results should be organized as json format:
> {Desire1: {Time: TIME, Desire: DESIRE, Actions: ACTIONS}}

Table A-3: Prompts for *desire* updating.

> **Prompts for desire update.**
>
> You are an virtual human in a room. Below is your profile:
> {VIRTUAL HUMAN PROFILE}
> Your previous desires are:
> {Desire 0:{DESIRE}}
> {Desire 1:{DESIRE}}
> ...
> Your current desire is {DESIRE}
> Your current desire can not be fulfilled in the room.
> The visible scene graph is {VISIBLE SCENE GRAPH}
> Please update your current desire. The output format should be json format: {Desire:DESIRE}

of detected objects and their properties. We provide an example of the input scene graph used in the LLM (Table A-6). *Belief* determines whether the *desire* can be fulfilled and the *intention* to be executed, as illustrated in Fig. 3 of the main manuscript. *Belief* is updated during exploration. We also provide the prompt design for exploration in Tab. A-7, where the LLM is prompted to determine a direction to explore based on the *desire* and the objects near the frontiers.

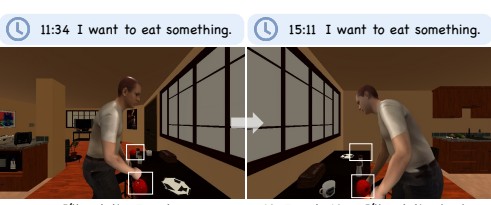

Figure A-3: The virtual human chooses to eat the Apple when the Tomato has been eaten.

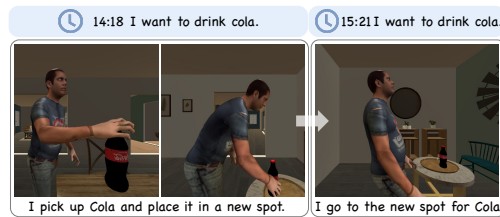

Figure A-4: The virtual human navigates to a different place due to Cola's location change.

Table A-4: Example of generated *intentions*.

---

**Example of generated intentions.**

*Desire: Take a short break and have a nutritious snack.*
**Intentions**: ["Navigate Apple_73", "PickUp Apple_73", "Navigate Sofa_3", "SitOn Sofa_3"]
*Desire: Take a cup of coffee.*
**Intentions**: ["Navigate Cup_33", "PickUp Cup_33", "Navigate CoffeeMachine_9"]

---

Table A-5: Prompts for motion style generation.

---

**Prompts for motion style generation.**

You are an virtual human in a room. Below is your profile:
{VIRTUAL HUMAN PROFILE}
Your previous intentions are {PREVIOUS INTENTIONS}. Your current intention is {CURRENT INTENTION}. Please generation a simple motion style description with only a few words. Example motion style: jogging, tiredly, briskly, run, naturally. Example answers: Example 1: A person runs. Example 2: A person walks tiredly.

---

Besides, *action* can also change the state of the environment, thus altering the *belief*. Consequently, the *scene dynamics* resulting from prior actions impact subsequent *intentions*. We illustrate this with two examples in Figures A-3 and A-4. In Figure A-3, the virtual human opts to eat the apple because the tomato has already been consumed. Similarly, in Figure A-4, the change in the cola's location prompts the virtual human to navigate to a different area.

## D   MORE RESULTS

We first analyze the effectiveness of our hand-object alignment strategy, and then provide more quantitative results to demonstrate the effectiveness of our method from the aspects of behavior diversity, motion quality, and exploration efficiency.

**Hand-object alignment.** As mentioned before, We propose a hand-object alignment strategy to generate more natural hand-object interaction poses. Fig. A-5 demonstrates the effectiveness of this strategy. According to the visualization, the baseline pose exhibits noticeable misalignment and causes the object to appear unnaturally floating. Our strategy successfully aligns the hand with the object geometry, yielding a significant improvement in interaction quality.

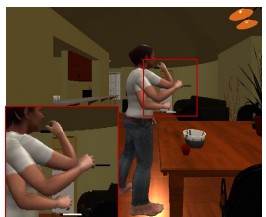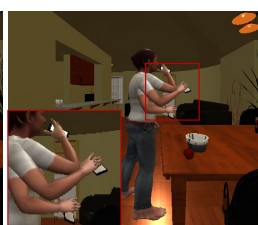
Before hand-object alignment          After hand-object alignment

Figure A-5: Effectiveness of hand-object alignment strategy.

**Behavior diversity.** To illustrate the diversity of virtual human behavior, we calculate the uniqueness ratio (I1/I2 Uniq.) and IoU of agent-object interactions under the same scene. As shown in Tab A-11, (a)-(e) represent different agent combinations within the same scene. The uniqueness ratio illustrates the overlap of objects interacted with by two individuals, demonstrating the high diversity of behavior generated by our framework.

Table A-11: Comparison of object interactions by different agents.

| Scene | (a) | (b) | (c) | (d) | (e) | Avg |
|---|---|---|---|---|---|---|
| I1 Uniq. | 77.8 | 75.0 | 50.0 | 80.0 | 77.8 | 72.1 |
| I2 Uniq. | 80.0 | 84.6 | 55.6 | 84.6 | 80.0 | 77.0 |
| IoU | 11.8 | 10.5 | 30.8 | 9.5 | 11.8 | 14.9 |

Table A-6: Example of *belief*.

Example of belief.

```
{
"AlarmClock_87": { "Location":[-3.5, 0.7, 3.3] },
"Apple_47": { "Location":[1.5, 1.0, -5.7] },
"BaseballBat_46": { "Location":[-2.7, 0.0, 2.6] },
"BasketBall_17": { "Location":[-0.9, 0.1, 3.3] },
"Bowl_27": { "Location":[1.7, 0.8, -4.5] },
"CellPhone_57": { "Location":[1.2, 0.6, 2.3] },
"Chair_54": { "Location":[1.4, 0.5, -1.7], "Affordance":Sit },
"Chair_55": { "Location":[1.9, 0.5, -0.8], "Affordance":Sit },
"Cup_39": { "Location":[-3.0, 0.9, -1.4] },
"Laptop_90": { "Location":[-4.2, 0.9, -2.3] },
"Bed_33": { "Location":[-3.2, 0.5, 2.1] },
"CoffeeMachine_120": { "Location":[1.7, 1.1, -5.4] },
... }
```

Table A-7: Prompts for exploration.

Prompts for exploration.

You are an virtual human in a room. Below is your profile:
{VIRTUAL HUMAN PROFILE}
Your current desire is DESIRE
Below is the objects around each direction, which direction should you explore?
Direction 1: {OBJECTS}
Direction 2: {OBJECTS}
...
The result should organized as json format: {Direction:DIRECTION, Reason:REASON}

**Analysis of SA-MDM.** As previously mentioned, we propose a hybrid spatial-aware motion representation that fuses joint positions with SMPL-X parameters. To evaluate this strategy, we apply joint-level control to the MLD trained on relative motion representation and conduct experiments across five human-scene combinations. As shown in Table A-12, while our action execution module still performs reasonably without global joint positions, control precision is compromised. The *Desire Exec.* drops significantly without global joint positions, and the *Goal Dist.* increases, highlighting the importance of global joint positions in improving the spatial awareness of the motion generation model.

Table A-12: Analysis of the influence of global joint position.

| Method | Desire Exec. ↑ | Action Exec. ↑ | Goal Dist. |
|---|---|---|---|
| w/o global (MLD) | 0.76 | 0.90 | 0.222 |
| with global | 0.89 | 0.94 | 0.205 |

**Human motion quality.** To demonstrate the quality of the generated human motion, we provide quantitative comparisons with OmniControl (Xie et al., 2024) on HumanML3D dataset (Guo et al., 2022). According to the results shown in Tab A-13, our method achieves lower FID and higher R-Precision, demonstrating the high quality of the generated human motion.

Table A-13: Results of motion generation on the HumanML3D dataset.

| Method | FID↓ | R-Precision (Top3)↑ |
|---|---|---|
| OmniControl | 0.31 | 0.693 |
| Ours | **0.27** | **0.773** |

Table A-8: Prompts for *intention* generation.

| Prompts for intention generation. |
| --- |

You are an virtual human in a room. Below is your profile:
{VIRTUAL HUMAN PROFILE}
You can use following actions and the objects in the scene graph to complete a desire:
[1] Open: open some articulated objects.
[2] Close: close some articulated objects.
[3] PickUp: pick up something, can be used for eat something or drink something.
[4] PutDown: put down something on something.
[5] Navigate: go to somewhere in the room.
[6] SitOn: sit on a chair or a sofa.
[7] LieOn: lie on a bed.
[8] StandUp: stand up from a chair, a sofa or a bed.
[9] DoIt: if the desire does not need interaction with the environment, just plan as "DoIt".
[10] Explore: explore the room if the current scene graph can not meet your intention.
Note: If the current scene graph can not meet your intention, Do not hypothesize anything. For example, you can not plan as "PickUp Something".
You should also be aware of previous intentions. For example, if the last previous action is "SitOn Chair_5", you should plan "StandUp Chair_5" before the next Navigation action.
Output Response Format: {"plan": task plan}
Ensure the output can be parsed as json format.
[Start Demonstration]
Demonstration 1:
Desire: I want to lie down.
Visible Scene Graph: {Chair_0, Chair_1, Cup_0, Cup_1, Table_0, Table_1}
Plan: {"plan": ["Explore"]}
Demonstration 2:
Desire: I want to drink some water.
Visible Scene Graph: {Cup_0, Cup_1, Sink_1, Chair_1, Chair_2}
Plan: {"plan": ["Navigate Cup_0", "PickUp Cup_0", "Navigate Sink_1"]}
Demonstration 3:
Desire: I want to sit to watch TV.
Visible Scene Graph: {Chair_0, Chair_1, Remote_Control_1, TV_1}
Plan: {"plan": ["Navigate Remote_Control_1", "PickUp Remote_Control_1", "Navigate Chair_1", "SitOn Chair_1"]}
Demonstration 4:
Desire: I want to sit down and stand up.
Visible Scene Graph: {Chair_0, Chair_1, Remote_Control_1, TV_1}
Plan: {"plan": ["Navigate Chair_0", "SitOn Chair_0", "StandUp Chair_0"]}
Demonstration 5:
Desire: I want to drink some water.
Visible Scene Graph: {Chair_1, Chair_2}
Plan: {"plan": ["Explore"]}
Demonstration 6:
Desire: I want to dance for a while.
Visible Scene Graph: {Chair_1, Chair_2}
Plan: {"plan": ["DoIt"]}
[End Demonstration]
Your current desire: {DESIRE}
Previous intentions: {PREVIOUS ACTIONS}
Visible Scene Graph: {VISIBLE SCENE GRAPH}

**Exploration efficiency.** To further demonstrate the effectiveness of *desire*-driven exploration, we compare it with the original frontier-based exploration approach. In addition to the planning success rate, we also calculate the average navigation distance (Nav. Dist.) required to reach the goal object. According to the results in Tab A-14, our *desire*-driven strategy enhances the planning success rate and significantly reduces the navigation distance.

Table A-14: Effectiveness of *desire*-driven exploration.

| Method | Daily Plan. ↑ | Desire Plan. ↑ | Nav. Dist.↓ |
|---|---|---|---|
| Frontier-based (Yamauchi, 1997) | 0.45 | 0.91 | 142.6 |
| *Desire*-driven | **0.65** | **0.95** | **94.9** |

# E DISCUSSIONS

## E.1 BROADER IMPACT

As virtual human agents advance, several societal and ethical considerations should be taken into account. These agents, capable of simulating human-like behavior, may influence social interactions by potentially reducing human-to-human contact, which could have implications for social dynamics, particularly in certain contexts or communities. Additionally, the use of data-driven approaches in developing these agents poses the risk of unintentionally reinforcing biases, especially if the training data does not sufficiently reflect the diversity of the real world. This could result in agents that inadvertently overlook or misrepresent certain groups. We hope that through our work, more people will recognize both the significance and challenges of autonomous virtual human agents, fostering greater awareness and discussion within the research community. By addressing key issues such as aligning agents' values with human values and improving the inclusiveness of training datasets, we aim to contribute to the development of virtual agents that are more ethically and socially responsible.

## E.2 STATEMENT OF LLM USAGE

In addition to implementing the "Mind", we also used LLMs to polish the manuscript. Specifically, we prompted LLMs to correct grammatical errors and improve the overall writing fluency.

## E.3 COMPARISON WITH OTHER COGNITION ARCHITECTURES

Behavior simulation has traditionally been approached through cognitive architectures such as SOAR (Laird, 2019), ACT-R (Anderson & Lebiere, 2014), and the BDI model (Rao et al., 1995). These frameworks have been instrumental in modeling reasoning and decision-making processes, particularly in the context of controlled cognitive tasks and psychological experiments. However, they largely abstract away the physical body and environment, and do not emphasize the continuous, dynamic interaction between cognition and physical experience.

In contrast, our proposed *EmbodiedHuman* architecture is grounded in the principles of *embodied cognition*, which emphasize that cognitive processes emerge through continuous interaction between mind, body, and environment. By integrating a structured cognitive module with motor execution and environmental feedback, our system constructs a closed-loop that allows virtual humans to perceive, act, and adapt in a physically grounded and context-sensitive manner. This embodied integration distinguishes our approach from prior cognition frameworks, and supports the development of virtual agents capable of more natural, context-sensitive, and adaptive behavior in complex environments. Tables A-15 and A-16 compare the actions generated by our EmbodiedHuman framework and SOAR, which we modified by incorporating an LLM and enabling environmental exploration and perception updates. The results demonstrate that our approach more accurately captures the value of virtual humans and produces actions that are both more reasonable and temporally continuous.

## E.4 EXTENSION TO MULTI-AGENT INTERACTION

Our framework is primarily designed for single-agent behavior modeling; nevertheless, it can be readily extended to multi-agent scenarios by incorporating *Interaction* as an additional *intention*. For example, when Fiona encounters Diego, Fiona will decide whether to initiate interaction, and Diego will decide whether to respond. If both agents choose to engage, as shown in Figure A-6, they begin the interaction by generating an interaction prompt for each agent (*e.g.*, "*talking to a person with gestures*").

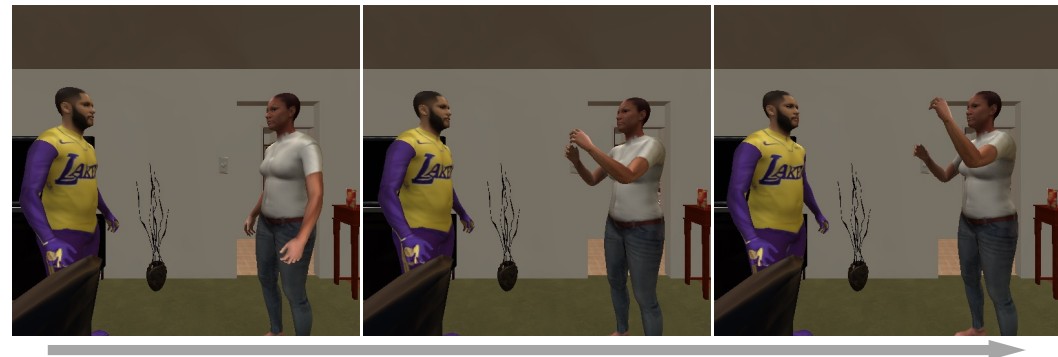

Figure A-6: An example of multi-agent interaction. Fiona and Diego talk with each other when they decide to interact.

### E.5 LIMITATIONS

Although our method enables complex interactions with the environment, there are still some limitations, as outlined below:

1) *Ultra-realistic motion quality.* Although our method supports free-form motion, certain actions requiring specific tools remain challenging to execute, such as "a person works out with a dumbbell." These actions are difficult to accomplish through joint control and may arise during *intention* generation, potentially leading to failures. Besides, due to the lack of full physical simulation, we can not achieve ultra-realistic human motion.

2) *Diverse multi-agent interactions.* Although our framework has the potential to be extended to multi-agent settings (as shown in Section E.4), it is currently limited to simple dialogue scenarios and cannot handle complex interactions, such as handshakes or collaborative task completion.

### E.6 FUTURE WORK

We identify three key directions to further enhance and expand the capabilities of our approach.

1) *Building an open-source ecosystem.* We will open-source our code and maintain it in the long term. Since our framework only requires an unknown scene and a virtual human profile as input, it is highly scalable. We will also continue to expand the benchmark. Finally, we will create a user-friendly interface to enable further research and development on this foundation.

2) *Enabling multi agent interactions.* Future work could further explore interactions among multiple virtual humans. One potential direction is leveraging an LLM to simulate social relationships between individuals and model their interactions—both verbal and physical—based on these relationships, which could also enable collaborative task completion. Besides, multi-human motion generation methods can be employed to enable complex interactive motions (*e.g.*, "hug", "shake hands").

3) *Improving motion quality.* In future work, it would be beneficial to incorporate physical information to enhance the quality of motion. This could involve leveraging imitation learning to achieve physics-based human motion control and incorporating additional physical attributes for perception, such as object weight.

Table A-9: Virtual human profiles: Part I.

**Diego**

**Profession**: Athlete
**Age**: 27
**Personality**: Diego is a dedicated athlete who places a strong emphasis on his fitness. His rigorous training routine is complemented by a balanced, high-protein diet that fuels his high-performance workouts. To stay sharp and energized throughout his day, he relies on his love for coffee, savoring each cup to maintain focus and alertness. After a long training session or busy day, Diego unwinds by occasionally enjoying a drink, using it as a way to relax and decompress. He maintains a strong sense of balance between his passion for fitness and his need for rejuvenation.
**Daily Goal**: Diego needs to do some exercise today.

**Fiona**

**Profession**: Manager
**Age**: 28
**Personality**: Fiona is a dynamic and outgoing manager at a thriving trading company, known for her exceptional leadership and strong interpersonal skills. With a natural talent for building relationships, she's a social butterfly who thrives in fast-paced environments, easily connecting with colleagues, clients, and partners alike. Her enthusiasm and positivity are contagious, making her a beloved figure in the office. Outside of work, Fiona has a deep passion for dancing. Dancing allows her to express herself creatively and unwind after a busy day.
**Daily Goal**: Today is a holiday.

**Rachel**

**Profession**: Dancer
**Age**: 27
**Personality**: Rachel is a vibrant and expressive dancer whose passion for movement flows into all aspects of her life. She loves experimenting with different cuisines but prioritizes a balanced diet rich in fresh vegetables, lean proteins, and the occasional indulgent dessert to keep her energy high for performances. Outside of dance, Rachel enjoys spending quiet moments practicing yoga or exploring art galleries for creative inspiration, always seeking beauty in movement and stillness alike.
**Daily Goal**: Rachel will hold a party tonight.

**Ben**

**Profession**: Engineer
**Age**: 27
**Personality**: Ben is a dynamic and outgoing engineer with a zest for life and a knack for solving complex problems. His charismatic personality shines through in both his professional and personal interactions. While he thrives on tackling engineering challenges, Ben also knows the importance of balance. He likes treating himself to some crispy fried food and a cold cola, especially after a productive day. With his blend of technical expertise and an easygoing nature, Ben embodies the perfect mix of dedication and fun.
**Daily Goal**: Ben needs to finish his project today.

**Sophia**

**Profession**: Researcher
**Age**: 27
**Personality**: Sophia is a passionate researcher dedicated to advancing the field of computer vision. Her work involves designing algorithms, analyzing complex visual data, and exploring the interplay between artificial intelligence and human perception. Outside her professional life, Sophia is a fitness enthusiast with a disciplined approach to health and wellness. She values physical activity as a way to maintain mental clarity and often engages in activities like yoga, running, or strength training.
**Daily Goal**: Sophia works from home today.

**Luke**

**Profession**: Student
**Age**: 26
**Personality**: Luke is a vibrant and curious student with a strong passion for geek culture. He loves immersing himself in the worlds of TV shows and video games, where he finds excitement, creativity, and endless entertainment. Whether it's unraveling intricate storylines or mastering new gaming levels, Luke thrives on intellectual challenges and imaginative experiences. His enthusiasm for pop culture and technology makes him a true geek at heart, always eager to learn, explore, and connect with others who share his interests.
**Daily Goal**: Today is a holiday.

Table A-10: Virtual human profiles: Part II.

| Clara | Armin |
|---|---|
| **Profession**: Saleswoman
**Age**: 32
**Personality**: Clara, a dynamic and personable saleswoman, has a knack for connecting with people and building lasting relationships. She enjoys staying active in her free time, often going for early morning runs to clear her mind before a busy day of meetings and client calls. With a preference for balanced, nutrient-rich meals, Clara usually starts her day with a smoothie packed with greens and fruit, keeping her energized and focused. In her downtime, she loves exploring local art galleries, finding inspiration in creative expressions outside her usual business-driven routine.
**Daily Goal**: Clara needs to make some phone calls today. | **Profession**: Programmer
**Age**: 24
**Personality**: Armin is a young programmer with a passion for solving complex problems and creating efficient, innovative solutions. Outside of work, Armin is curious and always eager to learn—he spends his free time exploring new technologies, reading about advancements in AI, or contributing to open-source projects. Despite his tech-heavy focus, he balances his life with hobbies like gaming, hiking, or playing chess, which help sharpen his strategic thinking. Socially, Armin is approachable and thoughtful, with a knack for explaining technical concepts in an understandable way.
**Daily Goal**: Armin needs to solve an issue today. |
| **Megan** | **Evan** |
| **Profession**: High school teacher
**Age**: 35
**Personality**: Megan is a thoughtful and dedicated high school teacher who finds joy in sparking curiosity and encouraging her students to reach their potential. She has a passion for cooking and enjoys experimenting with new, healthy recipes that she can share with family and friends. Outside the classroom, Megan loves hiking and exploring local trails, finding peace and inspiration in nature, and is also an avid reader, especially of historical novels and biographies that deepen her understanding of the world.
**Daily Goal**: Megan needs to prepare for a class. | **Profession**: Chef
**Age**: 38
**Personality**: Evan has a keen sense for balancing flavors, blending traditional techniques with modern twists to create dishes that are as visually stunning as they are delicious. With years of experience in diverse cuisines, Evan specializes in transforming fresh, locally sourced ingredients into mouthwatering meals that tell a story. Outside of work, he enjoys experimenting with new recipes, exploring global cuisines, and mentoring budding chefs, sharing his love for food and the art of cooking.
**Daily Goal**: Evan will try some new cooking. |

Table A-15: Generated actions of SOAR (Laird, 2019).

```
{
    "Action1": "Navigate Apple_115", "Action2": "PickUp Apple_115",
    "Action3": "SitOn Chair_38", "Action4": "PickUp Cup_39",
    "Action5": "PutDown Cup_39", "Action6": "StandUp Chair_38",
    "Action7": "Navigate Laptop_86", "Action8": "SitOn Chair_108",
    "Action9": "StandUp Chair_108", "Action10": "Navigate ArmChair_62",
    "Action11": "SitOn ArmChair_62", "Action12": "PickUp Newspaper_91",
    "Action13": "PutDown Newspaper_91", "Action14": "StandUp ArmChair_62",
    "Action15": "Navigate Bed_120", "Action16": "LieOn Bed_120",
    "Action17": "StandUp Bed_120", "Action18": "Navigate YogaSpot",
    "Action19": "DoIt: Yoga exercise", "Action20": "DoIt: Strength training",
    "Action21": "Navigate Sink_67", "Action22": "PickUp SoapBottle_40",
    "Action23": "PutDown SoapBottle_40","Action24": "Navigate Sofa_41",
    "Action25": "SitOn Sofa_41"
}
```

Table A-16: Generated actions of EmbodiedHuman.

```
{
    "Desire1": {
        "Time": "07:30",
        "Desire": "Start the morning with yoga to energize and clear my mind before
            work.",
        "Intentions": ["DoIt"]},
    "Desire2": {
        "Time": "08:10",
        "Desire": "Prepare a light breakfast and enjoy it while planning the day.",
        "Intentions": ["Navigate Apple_115", "PickUp Apple_115"]},
    "Desire3": {
        "Time": "08:45",
        "Desire": "Begin computer vision research and coding work on the laptop.",
            "Intentions": ["Navigate Laptop_86", "PickUp Laptop_86", "Navigate
            Chair_100", "SitOn Chair_100"]},
    "Desire4": {
        "Time": "10:20",
        "Desire": "Take a short coffee break to stay refreshed.",
        "Intentions": ["StandUp Chair_100", "Navigate Mug_30", "PickUp Mug_30",
            "Navigate CoffeeMachine_8"]},
    "Desire5": {
        "Time": "11:05",
        "Desire": "Continue working on algorithms and data analysis.",
        "Intentions": ["Navigate Chair_100", "SitOn Chair_100"]},
    "Desire6": {
        "Time": "12:40",
        "Desire": "Eat a healthy lunch with some bread and lettuce.",
        "Intentions": ["StandUp Chair_100", "Navigate Bread_58", "PickUp Bread_58"]},
    "Desire7": {
        "Time": "14:10",
        "Desire": "Have a midday stretch.",
        "Intentions": ["DoIt"]},
    "Desire8": {
        "Time": "14:35",
        "Desire": "Resume research tasks and writing code at the desk.",
        "Intentions": ["Navigate Chair_100", "SitOn Chair_100"]},
    "Desire9": {
        "Time": "17:15",
        "Desire": "Relax and read a book to wind down after work.",
        "Intentions": ["StandUp Chair_100", "Navigate Book_54", "PickUp Book_54",
            "Navigate Sofa_41", "SitOn Sofa_41"]},
    "Desire10": {
        "Time": "18:25",
        "Desire": "Prepare and enjoy a light dinner, drinking water to stay hydrated.",
        "Intentions": ["StandUp Sofa_41", "Navigate Cup_39", "PickUp Cup_39", "Navigate
            Sink_67",]},
    "Desire11": {
        "Time": "20:15",
        "Desire": "Unwind for the evening by lying on the bed and reflecting on the
            day.",
        "Intentions": ["Navigate Bed_120", "LieOn Bed_120"]}
}
```