# OpenReview forum: "Who and Where Am I? Embodied Cognition-Aware Virtual Humans"
_ICLR.cc/2026/Conference — ICLR 2026 Conference Withdrawn Submission_

### Official Review · Reviewer_Sdsu · 2025-10-21

**Soundness:** 2
**Presentation:** 4
**Contribution:** 3
**Rating:** 4
**Confidence:** 3

**Summary:**

The paper addresses the challenge of creating a virtual human that accurately simulates both internal mental states and external behaviors. It adopts the perspective of embodied cognition, wherein complex human behaviors emerge from dynamic interactions between the mind and the surrounding environment. The authors propose EmbodiedHuman, an end-to-end embodied cognitive architecture designed to naturally model human daily activities in indoor settings. To enable seamless motion and realistic human–scene interaction, the framework incorporates a spatial-aware motion diffusion model that generates plausible motion sequences conditioned on the scene context. Experimental results demonstrate that the proposed system is capable of exhibiting human-like, naturalistic behaviors in virtual environments, highlighting its potential for realistic embodied simulation.

**Strengths:**

The paper successfully constructs an end-to-end embodied cognitive architecture that plausibly interacts with virtual environments. It integrates high-level cognitive modeling with accurate motion simulation, aiming toward a holistic representation of human behavior.

The proposed system creatively models human–scene interactions by constraining a diffusion-based human motion simulator with a human–object penetration penalty, leading to more physically coherent and realistic movements.

The paper provides rich visualizations and clear system diagrams that make the complex architecture easy to follow, alongside vivid exemplar simulations that effectively illustrate the virtual human’s behavioral diversity.

**Weaknesses:**

The scientific significance of the proposed cognitive architecture is not fully convincing. The framework depends heavily on large language model (LLM) inference and motion diffusion models, which operate quite differently from actual human cognitive and motor processes. As a result, the system appears to approximate the outward appearance of human daily behavior rather than genuinely simulating human cognition. This raises doubts about both its practical utility and its relevance as a scientific model of human cognition.

Given the large scale and ambition of the proposed architecture, the evaluation scope appears narrow. Section 4.2 primarily provides qualitative examples with limited depth, and the core cognitive components, such as value, intent, and belief, are not explored in sufficient detail. Developing a human-like cognitive architecture should naturally lead to diverse avenues for comparative analysis with real human cognition, yet the paper largely focuses on quantitative assessments of individual components instead of broader cognitive insights.

The representation of human cognition seems oversimplified relative to its biological and psychological complexity. Human cognition involves intricate interactions among creativity, social relationships, memory, emotion, and other faculties. The proposed architecture abstracts these aspects into a small number of modules, with limited justification for the chosen decomposition. The overall impression is that the work emphasizes visible behavior and motion modeling over genuine cognitive understanding.

**Questions:**

Why is the embodied cognitive architecture structured specifically around these four cognitive modules? Are there concrete foundations or empirical evidence from neuroscience or cognitive science supporting this modularization? The introduction references theories from psychology and cognitive science, but could the authors clearly summarize these sources to justify the design? Additionally, where do broader cognitive functions such as creativity, social relation, or fear fit within these categories?

The proposed model appears to diverge from the mechanisms of real human cognition. Core modules such as value and belief are represented in explicit natural language and processed through a large general-purpose model (GPT) to rationally deduce actions. Similarly, motor control is simulated via a computationally expensive diffusion process that refines discrete motion frames, whereas human motor control is continuous and sensorimotor-driven.

If cognitive architecture departs substantially from the mechanisms of the human mind, what is the practical motivation for simulating human behavior in this form? If the system’s purpose is limited to imitating external appearance and normative behavior, it seems functionally similar to NPCs in video games. Could the authors clarify the intended real-world applications and the necessity of such a system beyond visual realism?

---

### Official Review · Reviewer_vox9 · 2025-10-27

**Soundness:** 2
**Presentation:** 3
**Contribution:** 2
**Rating:** 2
**Confidence:** 4

**Summary:**

This paper introduces a novel architectural framework, EmbodiedHuman, for simulating virtual humans. The framework integrates three modules: (1) a “Mind” module that enables embodied agents to reasoning about goals and intentions, supporting decision-making, (2) an “Action” module that translates decisions to motor actions in the simulated environment, and (3) an “Exploration” module that allows agents to explore their environment and learn by updating their mental states based on actions. The embodied agents (“virtual humans”) interact with common real-world objects (e.g., cups, food, devices, furniture) in a simulated environment, and plan actions based on environmental states and their own internal states. The paper introduces a spatial-aware motion diffusion model to generate body motions.

**Strengths:**

1. The paper is clearly written.

2. I believe the motion framework is original in its implementation and design.

**Weaknesses:**

1. The paper makes a passing reference to embodied cognition -- the theory in cognitive science that states cognition being driven by the need for goal-directed action -- without exploring that theoretical grounding. This reference is simply made to justify the claim that to simulate a virtual human we need to simultaneously simulate (1) beliefs and goals, (2) motor mechanics that may implement these goals, and (3) visual appearance. There is a missed opportunity here for grounding the framework in theories of human cognition.

2. It was unclear to me what does this model do that VirtualHome and Watch-And-Help does not do already?
I'd recommend clearly differentiating the contribution of the current work from other papers in the related work section.

3.
Generating realistic embodied human motion and behaviour is immensely hard, because you'd need to satisfy a huge number of motor constraints on muscles, balance, etc. Human observers are also very sensitive to minor motion cues, e.g. we can notice the slightest shifts of balance in an opponent about to score a penalty goal in soccer, to predict the direction the person will kick the ball.
For this reason, previous embodied human simulators like VirtualHome neither attempt to produce realistic motions, nor claim doing so as one of their goals.
In contrast, this paper makes an overstated claim that the model generates human-like motion and behaviour. The authors own video demonstrations (web-page) show that this motion is not realistic or human-like. I'd recommend toning-down this claim, to say that the generated motions functionally satisfies the agent's goals in the simulated environment, without claiming realism.

4. The evaluation appears to show scripted daily routines (e.g., eating, cooking, walking) that appear to largely sampled from a deterministic and environment-scripted list of goals. We are not shown any evidence that the system could handle more complex or open-ended tasks, which rises questions about the applications of this system.

5. Given that embodied cognition emphasizes relational and social context, the work misses an opportunity to explore richer multi-agent scenarios.

6. Realistic behaviour is claimed, but without a validation of this claim on human observers.

**Questions:**

1. How does the “Mind” module perceive the environment? Does it receive symbolic scene graphs, raw 3D states, or textual summaries from the simulator?

2. Is  LLMs desire generation curated in any way, to ensure the goals are meaningful, diverse, have a long-term value, and are achievable?

3. Can we see examples of failure cases? Do the agents get stuck in a loop, generate unfeasible their desires, repeatedly sample a small set of goals?

4. What are the applications of this system?

5. The paper claims that agents can learn, adapt and “evolve their behaviour,” but I did not see evidence of continual improvement or learning. I'd appreciate a clarification of this part.

---

### Official Review · Reviewer_VCwk · 2025-11-01

**Soundness:** 2
**Presentation:** 3
**Contribution:** 2
**Rating:** 4
**Confidence:** 4

**Summary:**

This paper presents EmbodiedHuman, a framework that attempts to bridge cognitive reasoning and embodied motion generation for virtual humans. The system combines a symbolic "Mind" model based on four causal variables (Value, Belief, Desire, and Intention\VBDI) with low-level embodied modules that control perception, exploration, and motion. At each step, a large language model (GPT-4o) updates the agent’s Desire and Intention variabls from textual prompts describing its situation, while a Spatial-Aware Motion Diffusion Model (SA-MDM) converts these intentions into physically plausible human motion within a 3D scene. The agent can also explore its environment through a frontier-based mapping procedure guided by its current Desire state.

In contrast to prior work such as TRUMANS, OmniControl, or TextSceneMotion, which focus on text- or scene-conditioned motion control, this paper introduces an explicit cognitive-reasoning layer that drives motion through internal mental variables rather than direct task or text supervision. The result is a simulated virtual human whose actions are produced through a loop between cognition (LLM-based reasoning), embodiment (motion diffusion and spatial control), and environment feedback. Experiments in Habitat 3.0 demonstrate qualitative examples of daily-life activities (e.g., eating, exercising, resting) and limited quantitative comparisons.

While the framework demonstrates a novel coupling between cognitive reasoning and embodied motion, the behaviors remain simple, and the work does not established scalability to more complex behaviors, the ability to maintain consistent goals and reasoning over time, or practical applicability beyond illustrative simulation.

**Strengths:**

- The paper tackles an important problem: connecting high-level cognitive reasoning with low-level physical motion (Modeling Value, Belief, Desire, and Intention) and linking them to action generation is an interesting step beyond standard motion-diffusion methods.

- The system is well structured, with separate components for cognition, perception, and motion. This makes the framework easy to understand and potentially useful for future extensions.

- The paper is well written and supported by clear figures that show how cognition connects to action. The visual examples help explain the framework’s purpose and operation.

- The paper points to a promising direction for embodied AI that combines reasoning, perception, and motion in a unified system.

**Weaknesses:**

1. The demonstrated behaviors are short, low-level activities (e.g., sitting, eating, picking up objects). The framework does not exhibit hierarchical or long-horizon planning, which is essential for realistic or adaptive human-like behavior.

2. The Desire and Intention updates rely entirely on handcrafted GPT-4o prompts. This makes cognitive reasoning heavily dependent on prompt design. The absence of structured reasoning rules or trainable components prevents reproducibility and limits scalability.

3. The framework lacks any mechanism for maintaining continuity in reasoning over time. Because the LLM has no persistent memory, the agent cannot recall or update prior mental states, resulting iin inconsistent reasoning and an inability to sustain coherent behavior. Prior work on memory-augmented or reflective LLM agents has shown that this limitation often leads to incoherent or repetitive reasoning [1, 2]. This absence of continuity also prevents the agent from tracking long-term roles or objectives, raising concerns about the framework’s applicability to modeling sustained human behavior.

4. The agent operates in isolation within static 3D environments. There is no modeling of communication, cooperation, or shared task execution among agents, which are central elements of embodied cognition.

5. The motion and exploration modules are well-engineered but largely extend existing approaches such as diffusion-based motion control and frontier mapping with minimal methodological innovation.

6. The quantitative evaluation is limited in scope and does not clearly islate the effect of the cognitive layer. Metrics focus on success rates or visual plausibility but omit measures of goal coherence, reasoning quality, or behavioral realism. There are also no user evaluations for the cognitive reasoning part.

[1] Noah Shinn et al.. Reflexion: Language Agents with Verbal Reinforcement Learning. NeurIPS 2023.
[1] Joon Sung Park et al.. Generative Agents: Interactive Simulacra of Human Behavior. UIST 2023.

**Questions:**

What concrete applications or downstream use cases are envisioned for this framework that integrate both the cognitive reasoning and action-generation components? In what scenarios would cognition-aware embodied agents provide clear advantages over existing simulation or motion-generation systems?

How does the framework scale to a broader set of human behaviors and tasks beyond the short daily routines demonstrated?

How is temporal context handled when updating the cognitive state through the LLM?
Is the model provided with the full sequence of past Value–Belief–Desire–Intention states and environment feedback, or only the current step?

What criteria are used to evaluate cognitive or behavioral success beyond visual plausibility of motion?

How might the proposed framework extend to multi-agent or interactive scenarios involving communication or shared goals?

---

### Official Review · Reviewer_MoxA · 2025-11-04

**Soundness:** 2
**Presentation:** 3
**Contribution:** 2
**Rating:** 4
**Confidence:** 3

**Summary:**

This paper presents EmbodiedHuman, an "embodied cognitive architecture" that includes Mind, a "structured cognitive module" that connects high-level cognition to low-level motor execution to simulate humans. Mind includes values, beliefs, desires, and intentions. EmbodiedHuman also involves an action execution model that turns intentions into embodied movements, and an exploration module that updates mental states through action feedback. Mind is based on an LLM, while the action execution module is based on a spatial-aware motion diffusion model.

The paper describes each component of EmbodiedHuman. Mind involves starting with selected values. It generates desires that drive exploration, builds up its beliefs using that exploration, uses its desires to get intentions, and then uses the action module to translate the intentions into actions. Values influence desire and intention, belief influences desire and intention, and desire influences intention.

The paper then goes through experiments and results. The first research question is "how does value influence virtual human behavior?" This is answered by showing examples of cases where the same desire with different values (e.g. the same desire of wanting food to satisfy hunger but with different values - enjoyment vs health) leads to different intentions and actions. The second research question asks the same but of environment rather than values; here, the same desire (wanting to chat) leads to different intentions due to different environments (in one case a cellphone is available, in another case a laptop is available). The paper also shows statistics demonstrating that plan following success rate drops without exploration, and that the action execution module beats baselines.

**Strengths:**

### Clarity
- Examples in results section are very useful - I often want to see more examples in main text in papers, and these are illuminating and build intuition
- RQ/module structure is helpful
- Writing is strong throughout the paper

### Quality
- The experiments that are there seem well-executed
- Design is compelling
- Action and Exploration module experiments are pretty good, though examples (like those in the main Mind module results sections) would help to understand why these two modules perform so much better than baselines.

### Significance and originality
To the best of my knowledge, this module is original.

**Weaknesses:**

### Quality
- It would help to have more discussion of the justification of the four-element Mind module structure. There are some references, but they aren't explained despite the audience being likely to be unfamiliar. Adding a clear discussion would help the paper.
- Related work lacking e.g. memory mechanisms and other crucial tools from LLM literature that may not already be used for human simulation, but are tools that supplement design elements contributed here.
- Results on the Mind module are weak. Though the examples are good,
  - There are some simple ablations, but there aren't exhaustive ablations. That would be crucial for such a complicated prompt-based system.
  - It's difficult to understand the robustness of the prompts/design with such limited comparison to baselines, ablations, etc.

This paper would be improved by significantly more, and more targeted, experimentation. Without that, it's highly ungrounded.

### Clarity
- Abstract is somewhat weak and flowery. It would help to have clear claims and statistics.
- At first, it's difficult to understand the _conceptual_ distinction from model-based reinforcement learning, as the conceptual language ("autonomously explore, plan, generate, and execute, with dynamic feedback that updates") isn't specific or differentiated. Later, I gather that the conceptual novelty is in the complexity of simulation and open-endedness. It would be godo to make that clear earlier.

### Significance and originality
The method's significance is buoyed by some of the results, but ultimately the Mind module is lacking in validation.

**Questions:**

- Where did the four-element Mind module structure come from?

---

### Note · Authors · 2025-11-12

I have read and agree with the venue's withdrawal policy on behalf of myself and my co-authors.